# Constitutive inflammation and epithelial-mesenchymal transition dictate sensitivity to nivolumab in CONFIRM: a placebo-controlled, randomised phase III trial

Leveraging adaptive tumour immunity to control mesothelioma via immune checkpoint blockade is now a standard therapeutic approach. However, the determinants of sensitivity remain elusive. Low non-synonymous mutation burden and programmed death-ligand 1 expression, an abundance of immunosuppressive immune cell infiltration, and 9p21 deletion should all mitigate responses to therapy. To address this knowledge gap, we conducted a double blind, placebo-controlled, randomized phase III trial of the PD1 inhibitor, nivolumab (ClinicalTrial.gov registration: NCT03063450). After 37.2 months of follow-up, the primary endpoint of progression free-survival, but not overall survival was met. The nivolumab response rate was 10.3%, and related grade 3 or above adverse events occurred in 20.4% versus 7.2% for placebo. Progression-free and overall survival were longer in nivolumab-treated responders *versus* non-responders. In an exploratory multiomic analysis, blinded whole exome, transcriptome and multiplex immune profiling were used to interrogate R- *versus* NR-subgroups. Non-synonymous and neoantigen mutation burden were no different between groups, however R-mesotheliomas were infiltrated with activated CD8[+] T- and CD19[+] B-lymphocytes, organised into tertiary lymphoid structures. B-cell infiltration correlated with pro-inflammatory chemokines including IL24 and CCL19. Conversely, epithelial-mesenchymal transition and mitosis were associated with resistance to nivolumab. These findings illuminate features which can be leveraged to advance precision immunotherapy in this rare cancer setting.

Malignant mesothelioma is a lethal cancer caused by asbestos associated with an average survival of 18 months[1]. It commonly arises from either the thoracic parietal pleura or, less frequently from the abdominal peritoneal lining. Despite recent advances, there remains a pressing, unmet need to identify new and effective treatments, particularly in the relapsed mesothelioma setting[2]. Over the last half decade, immune checkpoint blockade (ICB) has emerged as a new

therapeutic option[3]. Combined[4] or single agent ICB have shown clinical efficacy in patients with relapsed mesothelioma, however only a minority of patients respond, with resistance developing, on average by 24 weeks in the front-line treatment setting.

Mesotheliomas exhibit several features that are expected to mitigate responses to ICB. These include a low tumour mutation burden of around 2 mutations per megabase[5], low expression of

✉ e-mail: df132@le.ac.uk

programmed death-ligand 1 (PD-L1)[6], loss of human leucocyte antigen (HLA)[7], frequent 9p21 deletion[5,8], a markedly immunosuppressive tumour microenvironment[9,10], and epithelial-mesenchymal transition (EMT)[11]. The cellular and molecular features that dictate extreme response to ICB in mesothelioma are unknown.

Here we report the final efficacy results of the **C**heckp**OiN**t Blockade **F**or the **I**nhibition of **R**elapsed **M**esothelioma clinical trial (CONFIRM), a placebo controlled, double-blind randomised phase III clinical trial designed to investigate the efficacy of the anti-PD1 inhibitor nivolumab in patients with relapsed mesothelioma, and to uncover determinants of efficacy in patients exhibiting the most extreme response outcomes. This study was positive with respect to its co-primary endpoint of progression free survival. Blinded multi-omic analysis was employed to derive an explanatory model capable of forecasting mesothelioma response to immunotherapy.

## Results

### Patient Characteristics and treatment
From 10th May 2017 to 30th March 2020, 332 participants were enroled from 24 hospitals in the UK, of which 221 patients were randomised to receive nivolumab or placebo (111 patients) all of whom were included in the analysis (Supplementary Fig. 1). Co-primary endpoints were progression-free survival (PFS) and overall survival (OS). The median follow-up with censoring at death was 37.2 months inter-quartile range (IQR) 30.3–44.4. Baseline patient characteristics were balanced between groups, and all patients had received platinum-based therapy (supplementary note 1 tables 1A–C).

A median of 6 cycles (IQR: 3–13) of nivolumab without dose reductions, or 4 cycles of placebo (IQR: 3 to 8) were administered. At least one dose delay occurred in 96/217 (44.2%) of the patients who received nivolumab and 34/110 (30.9 %) in the placebo group (supplementary note 1 table 2). In all, 13/217 (6 %) of the patients in the nivolumab group and 5/110 (4.5%) of those in the placebo group completed protocol treatment. After discontinuation, 86/217 (39.6 %) of patients in the nivolumab group and 45/110 (40.9%) of patients in the placebo group received subsequent systemic cancer therapy (supplementary note 1 table 3). In the placebo group, 20/110 (18.1%) went on to receive immunotherapy following unblinding.

### Efficacy
The median investigator-reported and modified Response Evaluation Criteria in Solid Tumours (RECIST) reported PFS was longer for the nivolumab *versus* (*vs*) the placebo group (Fig. 1A, and supplementary note 1 table 4). Median investigator reported PFS was longer in the nivolumab treated group; 2.89 months (95% confidence interval [CI], 2.76 to 4.11; number of events 216/221), compared with 1.64 months (95% CI, 1.38 to 2.56; number of events 108/111) in the placebo group; unadjusted hazard ratio 0.65 (95% CI 0.51 to 0.82) supplementary note 1 table 4. The treatment effect for PFS favoured nivolumab for the pre-specified subgroups of patients with epithelioid histology (supplementary note 1 Fig. 2A-B).

PD-L1 tumour proportion score (TPS) assessed using 22C3 antibody was balanced between the treatment groups and was positive (>1%) in 88/258 (34.1%) of the randomised patients, with 73/258 (28.3%) scoring between 1–49%, and 15/258 (5.8%) >50%. It should be noted that the denominator evaluated was larger compared with previously reported in the preliminary CONFIRM trial report[6]. The PD-L1 negative rate i.e., <1% was 170/258 (65.9% supplementary note 1 table 5). PD-L1 TPS was lower in the non-epithelioid 9/26 (34.6%) *vs* the epithelioid subgroup 17/26 (65.4%). Significantly longer PFS was observed in the PD-L1 TPS > 50% subgroup (adjusted hazard ratio for the interaction was 0.28 (95% CI 0.09-0.94, p = 0.04), but not for either the >1% or 1–50% subgroups, supplementary note 1 and Figs. 3–4, table 5).

The analysis of overall survival was planned to coincide with 291 events. Patients who did not experience a survival event at this

threshold were censored at their completed end of study date. The median OS was 9.49 months (95% confidence interval [CI]; 7.69-11.01; number of events 198/221) *vs* 6.77 months (95% CI, 5.03-7.92; number of events 100/111) for the nivolumab *vs* placebo groups respectively, unadjusted hazard ratio 0.81 (0.64 to 1.04, p = 0.093 Fig. 1A, supplementary note 1 table 6). The adjusted HR was 0.81 (95% CI 0.64 to 1.04; p = 0.096). Restricted mean survival time analysis was conducted taking 12 months post-randomisation as the time point of interest. This showed that participants survived for 1.1 months longer on average (95% CI 0.2 to 2.0; p = 0.015) when receiving nivolumab. OS was not significantly longer for patients with PD-L1 > 50% (supplementary note 1 figure 6A–C, and table 7).

Objective response rate measured by modified RECIST was higher for the nivolumab group compared with placebo 20/195 (10.3%) *vs* 0/102 (0.0%). Some patients exhibited exceptional debulking of their mesotheliomas (Fig. 1B, and supplementary note 1 figures 5A-B). The response rate did not differ according to PD-L1 expression (supplementary note 1 table 8B). It should be noted that the modified RECIST response rate was re-computed centrally by the clinical trials unit, resulting a reduction of 1% from the investigator-reported 11% rate reported in the preliminary results publication[6]. Disease control rate was 79.5% (supplementary note 1 table 8A). The median time to partial response following nivolumab was 2.6 months (IQR 1.5 to 2.8), and the duration of response was 7.1 months (IQR 3.0-16.3) (supplementary note 1 table 8C–D).

### Tolerability and safety
Adverse events (AEs) classified as CTCAE4.0 grade 3 to 5 were similar between the nivolumab and placebo groups (supplementary note 1 table 9). The most common AEs leading to treatment discontinuation in the nivolumab arm were infusion related reactions, occurring in 30/221 (13.6%) people in the nivolumab group *vs* 3/111 (2.7%) in the placebo group. Serious adverse events were more frequent in the placebo group compared with the nivolumab group (supplementary note 1 table 10).

There were no treatment related deaths in either arm. Treatment related adverse events and treatment related serious adverse events, including haematologic and non-haematologic toxic events, occurred more frequently with nivolumab than placebo (supplementary note 1 tables 11–12). The median time to resolution of treatment-related adverse events was 7 days (IQR 1-28) in the nivolumab group and 5 days (IQR 1-5) in the placebo group. Median time to onset of pneumonitis affecting two patients was 84 days (IQR 42-126), with one event resolving in 2 days. There were no recurrences of pneumonitis.

### Somatic copy number amplification is enriched in nivolumab responders
Blinded multi-omic analysis was conducted on formalin-fixed paraffin embedded archival diagnostic tissue blocks (ie. prior to first-line therapy) in two subgroups corresponding to patients whose mesotheliomas exhibited either a partial response by modified RECIST (R, 16 patients) or progressive disease (NR, 13 patients, Fig. 1C–E, and supplementary note 1 figure 5C). PFS and OS in the R and NR subgroups is shown in Fig. 1F, G, note these Kaplan Meier curves are descriptive only and not formally comparable.

The genomic landscape of R- *vs* NR-subgroups was similar with respect to driver-specific copy number alterations, single nucleotide variants and uniparental disomy (Fig. 2A). Neither 9p21 nor *BAP1* alterations, which have been implicated as potential resistance or sensitivity predictors of ICB differed significantly between R- and NR-subgroups (Fig. 2B).

R-subgroup mesotheliomas exhibited a significantly higher frequency of amplified somatic copy number alterations compared with the NR-subgroup (Fig. 2C). However DNA damage response (DDR)

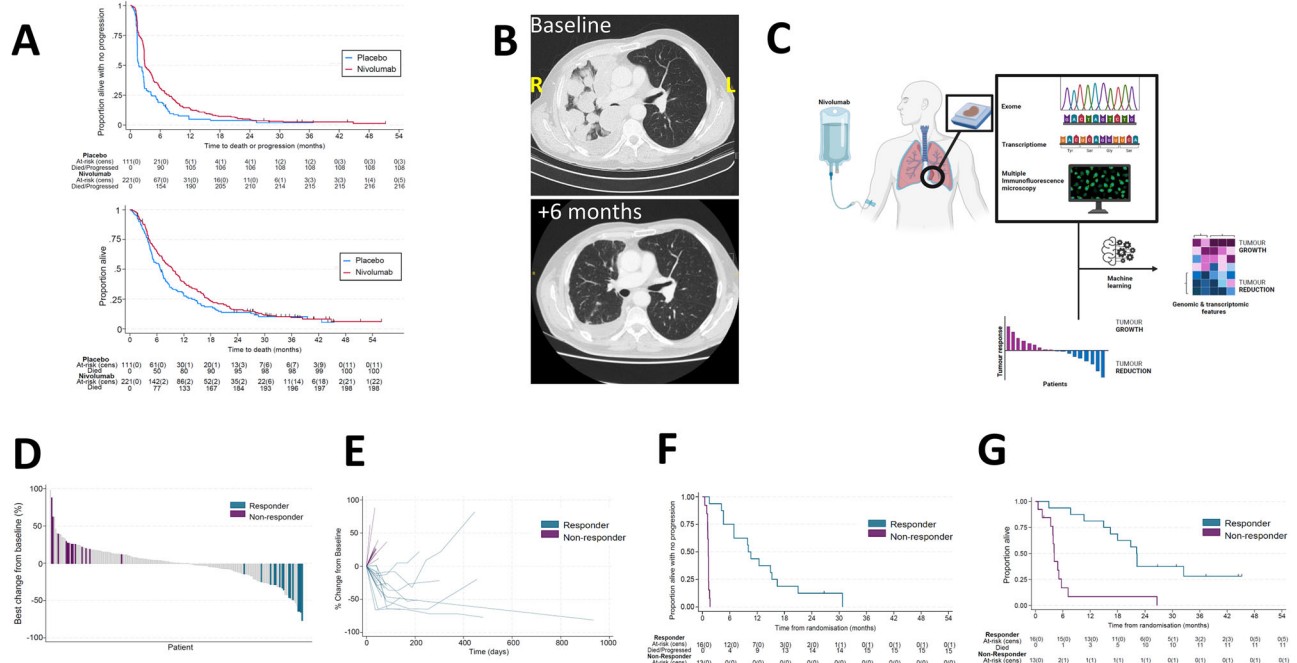

**Fig. 1 | Interrogating response heterogeneity via blinded multi-omic analysis of R- versus NR-mesotheliomas. A** Upper panel. Kaplan-Meier curve showing investigator-reported progression-free survival for nivolumab and placebo treatment groups in CONFIRM. Lower panel. Kaplan-Meier curve showing overall survival (proportion alive) for nivolumab and placebo treatment groups in CONFIRM. **B** Chest computerised tomography scan of the chest of a patient who exhibited an extreme response to nivolumab over the course of 6 months. **C** Archival formalin fixed tumour blocks corresponding to patients who received nivolumab and had either a partial response (R) or progressive disease (no response, NR) as their best outcome, were subjected to next generation sequencing of the whole exome or transcriptome. These tissues were also spatially phenotyped using multiplex-immunofluorescence microscopy. Created in BioRender. Fennell, D. (https://BioRender.com/sq9aqpm.) **D** Waterfall plot showing the selection of patients from R- vs NR-groups for blinded multi-omic analysis. **E** Spider plot showing the relative change from baseline for mesotheliomas in the NR- vs R-subgroups. **F** Kaplan-Meier plots for investigator reported PFS in the translational research cohorts denoted responder and non-responder respectively. These curves are descriptive only and are not formally comparable. **G** Kaplan-Meier plot for OS in the translational research cohorts denoted responder and non-responder respectively. These curves are descriptive only and are not formally comparable.

specific SCNAs did not show significantly higher enrichment in the R subgroup. *CHEK2* copy number losses were enriched in the R-group. However, tumour mutation burden (TMB), coding mutations (clonal or sub-clonal), predicted neoantigens (clonal or sub-clonal), or intra-tumour heterogeneity were similar between groups, as were the rates of somatic copy losses, gains and uniparental disomy (Fig. 2C, D, supplementary note 1 figure 7A-C). The frequency of gene fusions, which have been implicated as putative generators of neoantigen burden were analysed using two bulk RNAseq methods, ie. STAR-fusion and Arriba (supplementary note 1 figure 8A). No difference in the abundance of expressed fusions were found between between R- and NR-subgroups, although it should noted that the likelihood of degraded RNA could have impacted the sensitivity to call fusions (supplementary note 1 figure 8B).

### Pro-inflammatory transcription correlates with response to nivolumab
Deconvolution using gene set enrichment analysis, revealed significantly greater pro-inflammatory response-related transcription in the R-subgroup. Conversely, the NR-subgroup exhibited enrichment of EMT and mitosis-related signatures (Fig. 2E−G). EMT was associated with shorter OS (supplementary note 1 figure 9).

Significantly higher levels of multiple chemokines were expressed in R- *vs* NR- mesotheliomas, namely *CCL5, CCL16, CCL17, CCL19, CCL21, CCL25,* and *CXCL14* Benajmini-Hochberg adjusted *P* values < 0.05 (Fig. 3A). Expression of interleukins IL17C, IL23A and IL24 were greater in R- *vs* NR-mesotheliomas; Benajmini-Hochberg adjusted *P* values < 0.05 (Fig. 3B). B-lymphocyte infiltration inferred by transcriptome deconvolution, was correlated with both *IL24* and *CCL19* (Fig. 3C, D) and the ratio of *IL24* to EMT was significantly higher in R- *vs* NR- mesotheliomas

(Fig. 3E) and was associated for response with an AUROC of 0.889, $p = 0.001$ (Fig. 3F).

To explore the immune landscape of mesotheliomas in R and NR groups in CONFIRM In-silico immune-deconvolution was conducted using MCP counter. CD8 + T lymphocytes were significantly enriched in the R- *vs* NR-subgroup (Wilcoxon $p = 0.024$) Fig. 3G. CD8 + T cell deconvolution was orthogonally verified by multiplex immuno-fluorescence microscopy (ground truth) using an independent cohort of 98 mesotheliomas (Spearman's rank $r = 0.601$, $p = 0.602$). RNA deconvolution results were cross-validated using 3 alternative algorithms; quantiseq (Wilcoxon $p = 0.014$), EPIC (Wilcoxon $p = 0.026$), and ConsensusTME (Wilcoxon $p = 0.046$) supplementary note 1 figure 8C).

T-cell and B-cell receptor clonality and entropy did not differ significantly between R- and NR-subgroups (supplementary note 1 and figure 10).

### Tertiary lymphoid structures (TLSs) are enriched in nivolumab responders
R-subgroup mesotheliomas exhibited higher CD8 positive T cell and CD19 positive B cell densities assessed by multiplex immuno-fluorescence (Fig. 4A). Transcriptional enrichment of a germinal centre formation signature was enriched (albeit not significantly) in R mesotheliomas compared with the NR-subgroup (Fig. 4B). This was reflected morphologically by R-subgroup enriched TLSs, (Fig. 4C, D). TLSs were correlated with longer PFS and OS (Fig. 4E, and supplementary note 1 figure 11). TLS associated CD8 + T-lymphocyte and CD19 + B-lymphocytes were more abundant in the R- *vs* NR-group (Fig. 4D).

TLSs have been recently reported to be associated with antibody responses directed to expressed human endogenous retroviruses

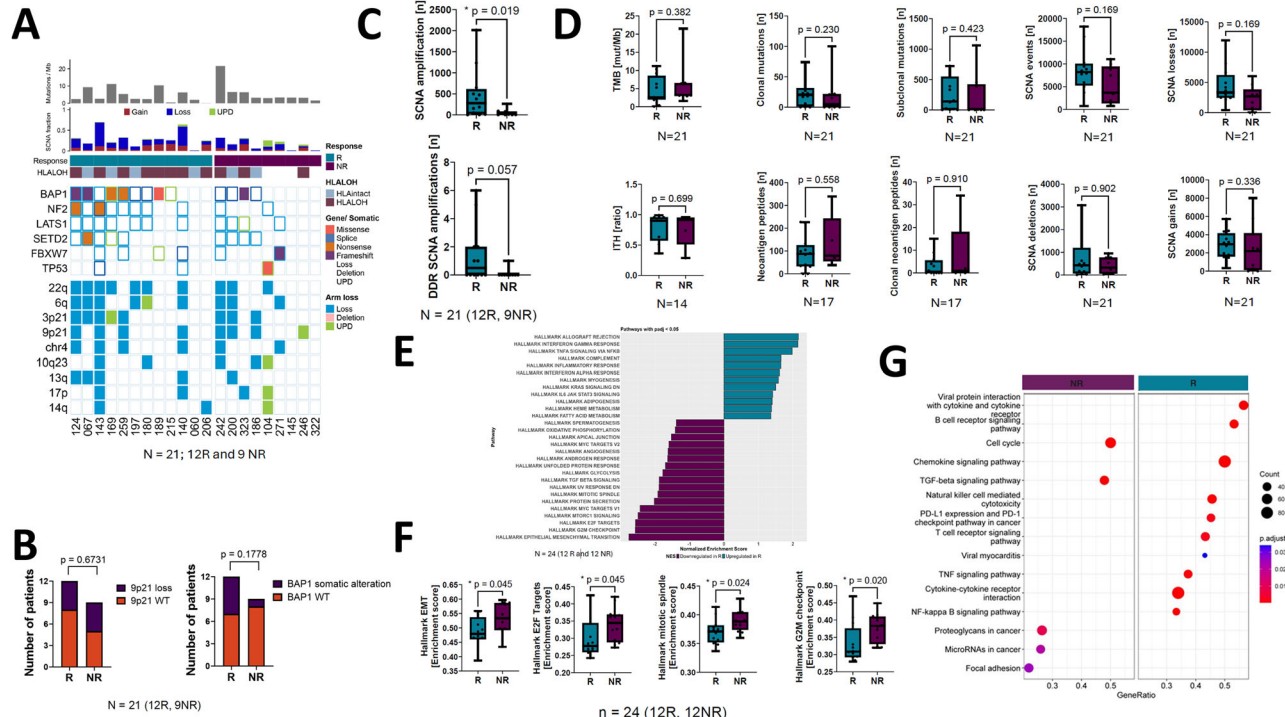

**Fig. 2 | Genomic correlates of response to nivolumab in the CONFIRM trial.**
**A** Heat map showing the relative burden of driver gene mutations or driver copy number alterations in the R- *vs* NR-subgroups. **B** Stacked histogram showing the relative somatic alteration frequency involving 9p21 (left) and BAP1/3p21 (right). BAP1 was not significantly enriched in R versus NR subgroups (NS, not significant). **C** Upper panel. Somatic copy number alterations (SCNAs) involving amplifications were more frequent in the R- *vs* NR-subgroups. Lower panel. Copy number alterations specifically affecting DNA damage response genes were not significantly different in the R- *vs* NR-subgroup. * indicates a Wilcoxon signed rank test two-sided *P* value equal to or less than 0.05. The boxplots show the median line and inter-quartile range (IQR, 25th-75th percentiles), with the whiskers extending to the maximum and minimum values. **D** Tumour mutation burden (TMB), clonal or subclonal mutations, intratumour heterogeneity (ITH), neoantigen burden (clonal or subclonal), or somatic copy number alterations (SCNAs). The boxplots show the median line and interquartile range (IQR, 25th–75th percentiles), with the whiskers extending to the maximum and minimum values. **E** Geneset enrichment analysis (GSEA) summary comparing R *vs* NR transcriptional signature enrichment. **F** Boxplots comparing geneset enrichment scores (single sample GSEA) in R- *vs* NR-subgroups. The boxplots show the median line and interquartile range (IQR, 25th–75th percentiles), with the whiskers extending to the maximum and minimum values. **G** Pathway enrichment plot showing inflammatory and chemokine pathway upregulation in R- *vs* NR-mesotheliomas.

(HERVs)[12], however, there was no significant difference in HERV expression between the R- and NR-groups (Fig. 4F).

## Discussion

In a proportion of patients suffering from mesothelioma, ICB can lead to dramatic tumour shrinkage with prolonged disease control and longer survival benefit. Presently, a combination of front-line anti-CTLA4 and anti-PD1 ICB is standard of care for all patients presenting with inoperable mesothelioma, however PFS remains short, averaging 6 months[4]. Nivolumab was initially licenced in the relapsed setting in Japan following the MERIT study (JapicCTI-163247)[13], and the aim of CONFIRM was to robustly determine the level of efficacy in a double-blind randomised placebo-controlled setting. Preliminary analysis showed significantly improved progression free and overall survival[6]. In this final analysis, the CONFIRM met its co-primary endpoint of progression free survival.

The advance of immunotherapy for patients with mesothelioma has been rapid[3], moving first from the relapsed setting to the front-line standard of care. Recently, it was reported that the addition of anti-PD1 ICB to standard chemotherapy is associated with an increase in overall survival compared with chemotherapy alone, by 6 weeks in the IND227 phase III clinical trial (NCT2784171)[14]. It is arguable that this modest benefit signifies a need for patient stratification using predictive biomarkers.

Response to systemic therapy in mesothelioma has been found to be associated with longer PFS in two reported meta-analyses[15,16]. We identified a subset of responding patients with mesothelioma

that exhibited significantly longer median PFS and OS compared to non-responders, then compared their genomic, transcriptomic, and immune phenotypes with that of PD1 ICB-refractory mesotheliomas. This cohort was admittedly small, comprising 29 patients in total. We consider this analysis of cellular and molecular correlates in this study, as hypothesis generating. Nevertheless, although these inferences were based on associative rather than mechanistic, distinct phenotypical features were found in the respective response phenotypes, leading a proposed model in which the balance of T- and B-cell inflammation involving TLSs in PD1 ICB sensitive mesotheliomas, contrasts with refractory tumours which are enriched for EMT, pro-mitotic transcription. Our findings are that these differences between response phenotypes exist in a largely similar genomic background.

In prE0505 (NCT02899195), a phase II trial evaluating chemotherapy with PD-L1 ICB, genomic instability and homologous recombination deficiency (HRD) was implicated as a putative predictive biomarker[17]. DNA damaging chemotherapy (platinum and pemetrexed) in prE0505 is probably sensitised in mesotheliomas with HRD. However, in CONFIRM, copy number amplifications harbouring DNA damage repair (DDR) genes were significantly enriched in nivolumab non-responders, whilst copy number loss of *CHEK2* was associated with response suggesting a possible role for DNA damage response deficiency in promoting sensitivity to ICB alone. Amplifications involving 1p have been reported to be associated with greater tumour inflammation, which may explain its association with nivolumab-responses[18].

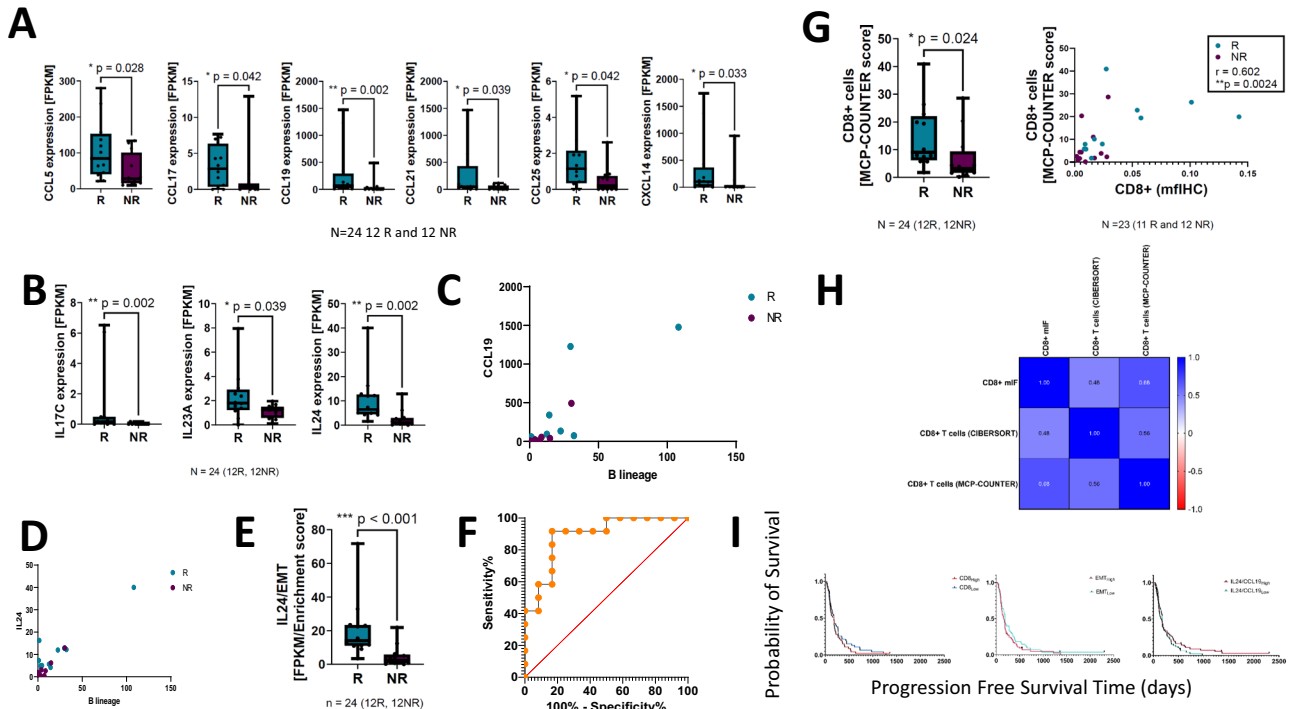

**Fig. 3 | Pro-inflammatory pathway transcriptionally upregulated in R- *vs* NR-mesotheliomas. A** Box plots showing relative chemokine expression corresponding to CCL5, CCL16, CCL17, CCL19, CCL21 and CCL23 transcripts in R- *vs* NR-mesotheliomas (* indicates a Wilcoxon signed rank test, two-sided *p*-value equal to or less than 0.05, ** equal to or less than 0.01). *Y*-axis units correspond to fragments per kilobase of feature per million mapped reads (TPM) normalised counts. The boxplots show the median line and interquartile range (IQR, 25th–75th percentiles), with the whiskers extending to the maximum and minimum values. **B** Boxplots showing Interleukins IL17C, IL23A and IL24 transcripts in R- *vs* NR-mesotheliomas. The boxplots show the median line and interquartile range (IQR, 25th–75th percentiles), with the whiskers extending to the maximum and minimum values. **C** Scatter plot showing CCL19 versus B lymphocyte lineage as determined by transcriptome deconvolution. Blue dots correspond to R-mesotheliomas and purple to NR (spearman's rank correlation coefficient= 0.85, *p* = 1.1 × 10⁻⁷). **D** Scatter plot showing IL24 versus B lymphocyte lineage as determined by transcriptome deconvolution. Blue dots correspond to R-mesotheliomas and purple to NR (spearman's rank correlation coefficient= 0.66, two-sided *p* = 0.0006). **E** Box plot comparing IL24/EMT ratio in R- *vs* NR- mesotheliomas (*** indicates a significance *p* < 0.0001). **F** Receiver operating characteristics plot for the ratio of IL24 transcript levels to EMT enrichment score in R- *vs* NR-mesotheliomas. AUROC was 0.9. **G** Left. Box plot showing the relative enrichment of CD8 immune cells in R (blue) *vs* NR (purple) mesotheliomas deconvoluted using MCP-Counter, Wilcoxon two-sided *p* = 0.024). This boxplot shows the median line and interquartile range (IQR, 25th–75th percentiles), with the whiskers extending to the maximum and minimum values. Right. Scatter plots showing CD8 + T cell enumeration by deconvolution versus multiplex immunofluorescence (mIF) respectively. Spearman's r = 0.6018, *p* = 0.002. **H** Correlation matrix showing a positive (blue) correlation between immune deconvolution algorithms (CIBERSORT, MCP Counter) and multiplex immunofluorescence for CD8 T cell enumeration. **I** Kaplan Meier curves showing progression free survival of CD8 T cells, EMT, or IL24/CCL19 ratio (dichotomised by the medians) in an independent (MEDUSA) cohort.

PD-L1 is a bona fide predictor of anti-PD1 efficacy in the lung cancer setting[19]. However, in patients with mesothelioma, the case for PD-L1 testing has been uncertain[4,20]. Here we show, that although PD-L1 TPS above 50% appears to be associated with longer PFS, the prevalence of this subgroup is low at around 5%, making PD-L1 TPS an unsuitable biomarker for patient selection.

EMT was found to be enriched in NR-mesotheliomas in CONFIRM, consistent with findings in both a phase 1 trial of atezolizumab and bevacizumab (NCT03074513)[21] and a phase II trial of pembrolizumab and nintedanib (NCT02856425), in patients with peritoneal and pleural mesothelioma respectively[22]. In a pan-cancer study, EMT was consistently associated with poor outcomes following immunotherapy[23,24]. EMT is a regulator of immune evasion, and in mesothelioma, exists as a continuum which correlates with sarcomatoid transformation. EMT is positively regulated by the cell cycle transcription factor E2F[25], whose activation signature was enriched in nivolumab-resistant mesotheliomas. Although upstream 9p21 deletion could potentially drive E2F via the p16ink4a-cyclin dependent kinase 4/6-Rb-cyclin E/CDK2 axis, we did not observe an enrichment of copy number loss at this locus in the NR-subgroup.

IL23 drives Th17 production of IL17 and can potentiate anti-tumour immunity in established cancers; both cytokines were found to be upregulated in R subgroup mesotheliomas suggesting a possible functional role for this axis in regulating sensitivity to PD1 inhibition in mesothelioma. The causes and potential therapeutic ramifications of IL23-IL17 signalling in mesothelioma warrants further investigation.

The cause of the observed constitutive inflammation and TLS formation in mesothelioma is unknown but suggests a persistent inflammatory stimulus arising from the time of diagnosis in the absence of an increased neoantigen burden in R- *vs* NR-mesotheliomas. The presence of TLSs has been reported to be a robust predictor of ICB efficacy across other cancers[26]. Failure to detect a higher neoantigen burden in R-mesotheliomas does not exclude the possibility that nivolumab responsive tumours harbour more immunostimulatory neoantigens. This idea is consistent with a report of robust CD8+ T clone activation by a mutant (but not wild-type) ROBO3P640H peptide in a patient with mesothelioma[27].

TLSs are associated with a favourable clinical outcome following ICB in several cancers[28–30] independent of PD-L1 expression, and are now being used as a putative predictive biomarker to prospectively stratify patients with solid tumours in the TAYLOR study (NCT05888857). Recently, machine learning has been applied to enable rapid detection of TLSs on routine histopathology images[31]. Such an approach could underpin precision-immunotherapy trials of the future, aimed at evaluating the predictive value of TLS-based patient stratification.

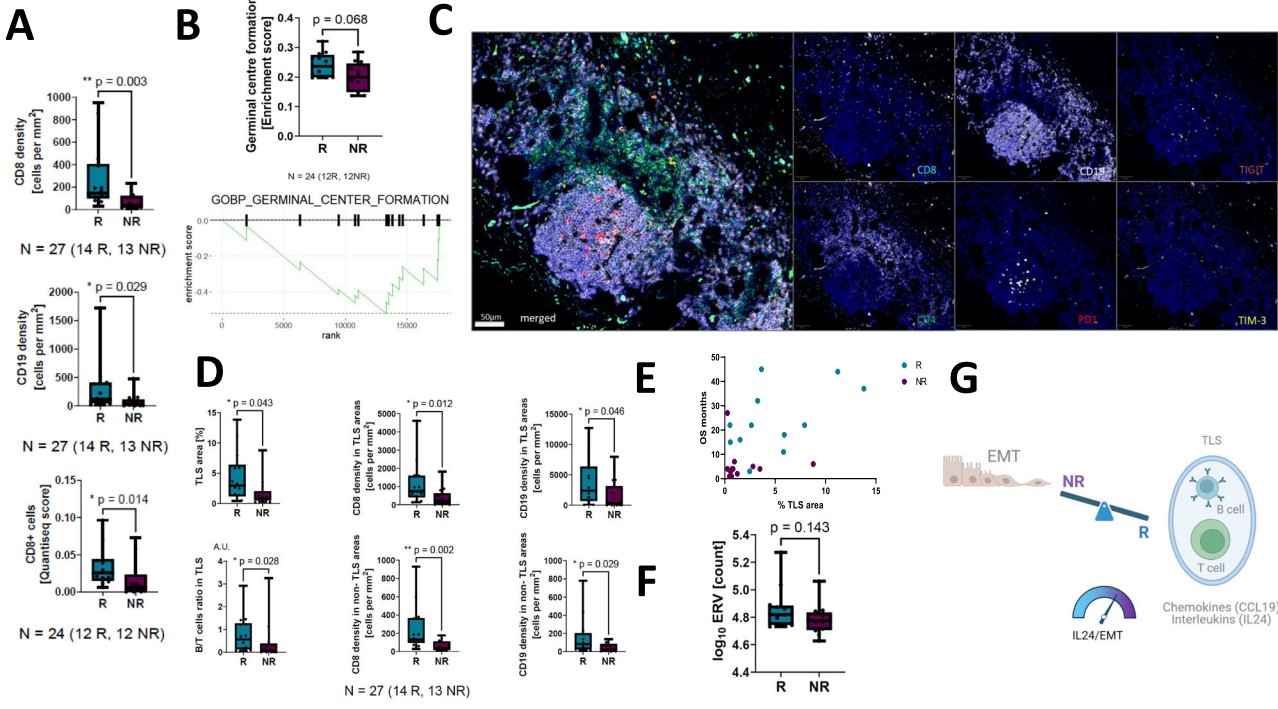

**Fig. 4 | Tertiary lymphoid structures are enriched in nivolumab responders.**
**A** Box plot showing a significantly higher number of infiltrating CD8+ T- lymphocytes in R-subgroup mesotheliomas compared with NR. The boxplots show the median line and interquartile range (IQR, 25th–75th percentiles), with the whiskers extending to the maximum and minimum values. **B** Geneset enrichment plot showing germinal centre formation transcriptional signature enrichment in R- vs NR-mesotheliomas. This boxplot shows the median line and interquartile range (IQR, 25th–75th percentiles), with the whiskers extending to the maximum and minimum values. **C** Micrograph showing a representative lymphoid aggregate (TLS) using a CD8, CD4 CD19, PD1, TIM-3, TIGIT multiplex immunofluorescence panel. **D** Box plots showing the relative TLS abundance in R- vs NR-groups with

respect to TLS area, T-cell density both within and outside TLS areas. B-cell density per TLS area and B-cell density within and outside of the TLS areas (* Wilcoxon signed rank test P value < 0.05, **p < 0.01). The boxplots show the median line and interquartile range (IQR, 25th–75th percentiles), with the whiskers extending to the maximum and minimum values. **E** Scatter plot showing the correlation between % TLS area and OS (r = 0.47, p = 0.02), R is shown in blue and NR in purple.
**F** Endogenous retroviral gene expression showed no difference between R- vs NR-subgroups (NS, not significant). **G.** Schematic showing the relative balance between stemness (EMT) and inflammation, reflected by chemokine expression and TLS formation. IL24:EMT ratio predicts response, PFS and OS. This figure was created using BioRender. Fennell, D. (https://BioRender.com/4dwlhk1).

The cause of TLSs in mesothelioma is presently unknown. A recent report from the TraceRx consortium identified HERVs as drivers of TLS formation and immunotherapy response in non-small cell lung cancer[12]. In CONFIRM however, HERV expression was not found to correlate with response to PD1 ICB. Tumour extrinsic factors such as the gut microbiome have been reported to forecast response to PD1-ICB through the intra-tumour regulation of T- and B-cell infiltration[32]. Furthermore, intra-tumour microbiota appear to exhibit tumour specific tropism and can influence the response to immunotherapy[33]. In the absence of a clear genomic signal to account for immune infiltration in mesothelioma, the gut microbiota represents a potential modifier of ICB in patients with mesothelioma, as recently suggested in the MIST4 phase II clinical trial (NCT03654833)[34].

Taken together, our findings suggest that ICB is most effective in inflamed mesotheliomas harbouring high *IL24/CCL19* expression, TLSs and low EMT (Fig. 4G). These findings suggest that a precision medicine approach to treat mesothelioma patients is warranted to maximise patient benefit, and should be considered in future biomarker-guided clinical trial designs.

## Methods
### Ethics statement
The study protocol was approved by the West Midlands, Edgbaston Research Ethics Committee (16/WM/0472). Further analysis of formalin fixed tissue was conducted under research ethics approval 14/LO/1527, a translational research platform entitled Predicting Drug and Radiation Sensitivity in Thoracic Cancers. This research platform was

approved by the University Hospitals of Leicester NHS Trust (reference IRAS131283 and 14/EM/1159) with the University of Leicester being a sponsor. The study was completed in accordance with the provisions of the Declaration of Helsinki and Good Clinical Practice guidelines as defined by the International Conference on Harmonisation. Written informed consent was obtained from all patients before enrolment.

### Study design and participants
CONFIRM was a multicentre, double-blind, placebo-controlled, parallel group, randomised phase 3 trial design (NCT NCT03063450). Enrolment involved 24 hospitals in the UK. Patients were eligible if they were aged 18 years or over with histologically confirmed pleural or peritoneal mesothelioma of any histological subtype, and who had been previously treated with at least one course of standard of care, chemotherapy with radiologically confirmed progression. Patients required an Eastern Cooperative Oncology Group (ECOG) performance status score of 0 or 1, radiologically assessable disease according to modified Response Evaluation Criteria in Solid Tumours (RECIST) or RECIST version 1.1, and an archival tumour biopsy for blinded multi-omic analysis.

The following laboratory criteria were mandated: a white blood cell count of at least $2 \times 10^9$ cells per L, neutrophil count at least $1 \cdot 5 \times 10^9$ cells per L, platelet count at least $100 \times 10^9$ per L, haemoglobin concentration at least 90 g/L, serum creatinine concentration of up to $1 \cdot 5 \times$ the upper limit of normal (ULN) or creatinine clearance higher than 50 mL/min (using the Cockcroft-Gault formula). Liver function tests i.e., aspartate aminotransferase concentration was allowed up to

3 × ULN or alanine aminotransferase concentration up to 3 × ULN (if both are assessed, both needed to be up to 3 × ULN), and total bilirubin concentration up to 1·5 × ULN (except patients with Gilbert syndrome, who had to have total bilirubin <51.3 μmol/L).

Patients were approached in the hospital setting by research staff. There was no restriction on the number of previous therapies received. Key exclusion criteria included previous treatment with an immune checkpoint inhibitor, uncontrolled metastasis involving the CNS, and autoimmune disease. The complete eligibility criteria are provided in the study protocol (supplementary note 2). Median survival with no additional treatment was expected to be ~6 months for eligible patients.

### Randomisation and masking

Patients were randomised in a 2:1 ratio to receive either nivolumab or placebo with an interactive web response system after the recording of baseline measures completion of screening. The randomisation sequence was generated with Alea. Patients were stratified by histology (epithelioid vs non-epithelioid), with random block sizes of 3 and 6. Treating clinicians and participants were masked to group allocation, but unmasking could be requested by the treating clinician following disease progression.

### Treatment

A flat dose of 240 mg of nivolumab or placebo (saline) was administered intravenously over 30 min every 2 weeks and was continued until either radiologically confirmed disease progression, withdrawal from treatment, or for a maximum of 12 months. The 12-month cap was justified due to the expected magnitude of both progression-free survival (PFS, median 3 months) and overall survival (OS median 6 months) durations, which are short for patients with mesothelioma in the relapsed treatment setting. This pragmatic cap was agreed with the supplier of nivolumab, Bristol Myers Squibb. Dose delays were permitted for up to 4 weeks from the previous dose.

Criteria for dose delay included any grade 2 non-skin, drug-related adverse events; any grade 3 skin drug-related adverse events; and any grade 3 drug-related laboratory abnormality adverse events. Treatment interruptions were permitted. Re-commencement of the infusion was recommended for grade 2 symptoms, but discontinuation was recommended for grade 3 or greater adverse events. Full requirements for treatment delay or discontinuation because of treatment-related adverse events are specified in the trial protocol. Reduction in the treatment dose was not permitted.

Computerised tomographic (CT) scans were reviewed locally, not centrally. Adverse events were assessed on day 1 of each cycle, 4 weeks following treatment discontinuation, and up to and including 100 days after treatment discontinuation. Grading used the National Cancer Institute Common Terminology Criteria for Adverse Events version 4.03. Laboratory parameters (serum chemistry, full blood count, liver function tests, and thyroid function tests) were assessed on day 1 of each cycle until disease progression and 4 weeks after treatment discontinuation.

### PD-L1 tumour proportion score

Retrospective evaluation of PD-L1 protein expression employed pretreatment tumour-biopsy specimens, and employed a validated, automated immunohistochemical assay with a rabbit monoclonal anti-human PD-L1 antibody (clone 22-C3) according to guidelines laid out in accordance with PD-L1 immunohistochemistry 22C3 pharmDx (Agilent, Santa Clara, CA, USA). Evaluation was independently validated by a consultant histopathologist and advanced biomedical scientist. Tumour cell membrane positivity for PD-L1 expression was observed at a prespecified expression threshold of <1%, 1–50% of cells, and >50% of cells in a section that included at least 100 evaluable tumour cells.

### Clinical outcomes

The co-primary endpoints were investigator reported PFS (hereafter referred to as PFS) as the time from randomisation to disease progression according to blinded investigator assessment or death, whichever occurred first, and OS *i.e.*, the time from randomisation to death from any cause. Co-primary endpoints were monitored every 3 months following discontinuation of treatment.

Secondary endpoints were overall response to treatment, defined as either complete or partial response, stable disease, or progressive disease all determined by mRECIST or RECIST 1.1; 12-month OS and PFS; safety; and efficacy (for PFS and OS) according to tumour PD-L1 tumour proportion score. Quality of life (EQ-5D) and cost per QALY data were collected as part of the trial and will be reported in a separate publication.

### Statistical analysis

All statistical analyses were done with Stata (version 16.0). Sample size was based on a hazard ratio (HR) of 0·7 for OS (equivalent to an improvement in median OS from 6·0 to 8·5 months), 80% power, 4 years of recruitment, and 6 months of follow-up. This led to a target sample size of 336 participants (291 events). A two-sided α of 0·04 was chosen based on interim analyses for efficacy for OS. One formal interim analysis for futility was carried out after 74 (25%) OS events had occurred in June 2019, (median follow up 5·09 months (IQR 3·91–6·90). The study continued as planned after this interim analysis.

Almost 2 years into recruitment (Feb 14, 2019; protocol amendment 6), PFS was added as a co-primary endpoint due to concerns that immunotherapy might be increasingly used off-study following progression, thus affecting the estimate of the effect of nivolumab on OS. Following the addition of PFS as a co-primary endpoint, an α of 0·04 was maintained for OS, based on a hierarchical testing procedure, designed to maintain an overall α of 0·05 across the co-primary endpoints. This procedure used two α values to determine significance for PFS depending on whether OS was significant (α 0·05) or not (α 0·01). The sample size of 336 participants gave more than 80% power for a HR of 0·65 for PFS (with α 0·01). This change was approved by the independent Trial Steering Committee and was included in a protocol amendment 6.

On Jan 13, 2020, it was agreed, and approved by the independent Trial Steering Committee, that the preplanned interim efficacy and futility analyses should be removed (~3 months before the anticipated end of recruitment; protocol amendment 7; June 11, 2020). The efficacy analysis was based on PD-L1 status, and recruitment was almost complete once sufficient samples were obtained and analysed. Interim futility analyses were removed due to them being done too near to or after the end of recruitment (as a consequence of faster than anticipated recruitment), restricting their perceived value. No other modifications were made to the study.

Investigator-reported PFS and OS were analysed with a Cox proportional hazards model, adjusted for epithelioid type (because this was a stratification factor). Significance thresholds were 0·04 for OS, and either 0·05 if OS was significant or 0·01 if OS was not significant for PFS. Survival curves for each group were estimated with the Kaplan-Meier method, and non-proportionality was assessed visually. Survival rates were derived from the Kaplan-Meier estimates. Prespecified sensitivity analyses were done to evaluate the association of pre-study status with respect to PD-L1 expression defined as either positive or negative, using a group by expression interaction term in the Cox model. Median time to onset of treatment-related adverse events and median time to resolution of treatment-related adverse events were assessed in a post-hoc analysis using the observed median time.

Both co-primary and secondary efficacy analyses and safety analyses included all patients who were randomly assigned. The only exception was for the PD-L1 analysis, for which only patients with assessable tissue samples were included. Analysis was based on the

treatment policy estimand, in which patients were analysed according to the group they were randomly assigned to and regardless of other treatments, such as off-trial immunotherapy (equivalent to the intention-to-treat principle). A prespecified analysis of PFS and OS across prespecified baseline characteristics with forest plots was done. Median time to response and duration of response were included as post-hoc analyses. A prespecified efficacy analysis by PD-L1 subgroups were assessed for PFS and OS.

The difference in restricted mean survival time between groups was calculated for overall survival as a post-hoc analysis, using the *strmst2* command in Stata. The 12-month time point was chosen as this represents the maximum length of treatment.

With respect to statistical analysis of the multi-omics cohort comparison of R and NR related variables employed the non-parametric Wilcoxon signed rank test. Given that the sample size was small, a permutation test was applied to estimate the Wilcoxon test statistic and *P* value. The data was split into two groups; Responders (R) and Non-Responders (NR). R (version = 4.3.2) was used to compute and permutate the data. The observed data points were randomly shuffled (N permutations=1000) for the R versus NR group comparisons to create new permutations. For each of the N permutations, the shuffled R and NR groups were used to calculate the Wilcoxon W' (W prime) statistic and stored in a list. The permuted p-value was determined by calculating the proportion of permuted statistics greater than the observed statistic, divided by the total number of test statistics calculated, which is equal to the number of total permutations (n), as seen in (1), ie.

$$p = \frac{\sum (W' \geq W)}{n} \tag{1}$$

For bi-variate correlation analysis, non-parametric analysis was computed using the Spearman's rank correlation coefficient. Prism version 9.5.1 (Graph, San Diego, CA, USA). Illustrations were created with biorender.com.

## Formalin fixed paraffin embedded (FFPE) tissue assessment and processing

Diagnostic formalin fixed tissue blocks were utilised for correlative studies (supplementary note 1 and figure 6). These were collected from the 24 UK treatment centres, and subsequently sectioned to generate Haematoxylin and Eosin (H&E) slides which were examined by a histopathology advanced biomedical scientist with support from a consultant histopathologist, who identified and marked representative areas of tumour on the H&E stained slides from the FFPE tumour blocks. Multiple tissue cores (1.0 mm each in size) were taken from the marked areas. DNA was isolated from these tissue cores using the MagMAX™ FFPE DNA/RNA Ultra Kit (ThermoFisher Scientific, Waltham, MA, USA #A31881) on the Kingfisher™ Flex sample purification system (ThermoFisher Scientific, Waltham, MA, USA) as per manufacturer's instructions. DNA was quantified using the Qubit™ 1 x dsDNA HS assay (ThermoFisher Scientific, Waltham, MA, USA #Q33230) on the Qubit™ 4.0 fluorometer according to manufacturer's instructions.

## Germline DNA extraction

Blood samples were collected in EDTA tubes and blood components (buffy coat and plasma) were isolated ≤2 h post-venepuncture. Whole blood was firstly centrifuged for 10 min at 1000 × *g* at 4 °C to separate blood components. Plasma was carefully transferred to a clean tube, avoiding visible contamination from the buffy coat and packed erythrocytes, and then centrifuged for a further 10 min at 2000 ×s *g* at 4 °C to exclude any residual cellular matter, before finally being aliquoted into 1.5 mL tubes. Both centrifugation steps were conducted with the centrifuge brake set to zero. Leucocytes, a source of germline control DNA, were isolated by carefully pipetting the buffy coat layer

and transferred to 1.5 mL tubes. Both plasma and buffy coat aliquots were placed at −80 °C for later use. Germline DNA was isolated from buffy coat using the QIAamp DNA Blood Mini Kit (Qiagen, Hilden, Germany 51104). DNA was quantified using the Qubit™ 1 x dsDNA HS assay (ThermoFisher Scientific, Waltham, MA, USA #Q33230) on the Qubit™ 4.0 fluorometer (ThermoFisher Scientific, Waltham, MA, USA).

## Whole exome sequencing (WES)

The whole-exome sequencing library was prepared from 1 μg genomic DNA by using the Agilent SureSelect Human All ExonV6 kit (Agilent Technologies, San Diego, CA, USA). Index codes were added to each sample. DNA fragmentation (180-280 bp) employed hydrodynamic shearing system (Covaris, Wobum, MA, USA). Remaining overhangs were converted into blunt ends via exonuclease/polymerase and enzymes removed. After adenylation of 3′ ends of DNA fragments, adaptor oligonucleotides were ligated. DNA fragments with 3′ and 5′ ligated adaptor molecules were enriched by polymerase chain reaction (PCR), followed by hybridisation with a biotin labelled probe. Streptomycin-coated magnetic beads were then used for exon capture, which were then enriched by PCR to add index tags to prepare for hybrization. Purification employed the AMPure XP system (Beckman Coulter, Brea, CA, USA). Quantification employed the Agilent high sensitivity DNA assay on the Agilent Bioanalyzer 2100 system (Agilent Technologies, San Diego, CA, USA). Qualified exome capture libraries were then sequenced on Illumina NovaSeq 6000 platform (Illumina, San Diego, CA, USA), according to standard protocols, for 150 bp paired-end multiplexed sequence. After sequencing, mean coverage of tumour and normal exomes were both 276X.

## Processing of WES data

After removing sequencing reads with low quality and adaptor bases using FASTP, clean reads were aligned to human reference genome (UCSC hg19) using Burrows-Wheeler Aligner (bwa-0.7.17). Mapped genomes were sorted using Sambamba (v0.6.7). Duplicate reads were marked using Picard tools (v2.18.9). Somatic SNVs and INDELs were detected with VarScan2 and MuTect2 jointly. Briefly, VarScan2 somatic (v2.3) were used to do somatic variants calling with default parameters, except for the following: minimum coverage for normal and tumour sample were set to 10 and 8 separately, minimum variant frequency was adjusted to 0.01 and tumour purity was set to 0.5. As to MuTect2 dealing process, we used MuTect2 contained in GATK bundle (4.0.5.1), with default parameter. ANNOVAR was used for functional annotation of variants. For germline DNA, no correction for clonal haematopoiesis was conducted.

## SNV and INDEL calling

Somatic SNVs and INDELs were detected with VarScan2 and MuTect2. Briefly, VarScan2 somatic (v2.3) were used to do somatic variants calling between tumour and matched normal samples based on the output from SAMtools mpileup (1.0). Default parameters were used except for the following; minimum coverage for normal and tumour sample were set to 10 and 8 separately, minimum variant frequency was adjusted to 0.01 and tumour purity was set to 0.5. After which VarScan2 processSomatic was used to extract somatic variants with minimum tumour frequency 0.01 and maximum normal frequency 0.05. Then bam-readcount (0.8.0) and Varscan2's wrapped fqfilter.pl were combined to conduct the mutation filtering. We used MuTect2 contained in GATK bundle (4.0.5.1) with default parameters.

All detected variants were annotated with Annovar (14 Dec 2015). Main databases used in future filtering or downstream analysis are as follows: SIFT, PolyPhen and MutationTaster scores used to predict the deleteriousness of mutations; Alternative allele frequencies in populations reported by large scale sequencing projects 1000 Human Genome (1000 G), Exome Aggregation Consortium (ExAC) and exome

sequencing project (ESP); Other databases including dbSNP, COSMIC, GO and KEGG.

## Minimisation of false variant calls

To reduce false positive variant calls, further filtering strategies were used on the mutation detection results of both MuTect2 and VarScan2. An SNV would be considered a true positive call if it satisfied the following conditions:

**Variant allele frequency.** The SNV was called by MuTect2 and VarScan2 (somatic $P$ value $\leq 0.1$) simultaneously, and both with a variant allele frequency (VAF) no < 2%; or only detected by VarScan2, but with a VAF greater than 5%.

**Variant allele frequency in normal samples.** VAF in matched normal samples for the position need to be <1% and reads number for alternative alleles is <5.

**Blacklist filter.** The SNV was not located in the blacklist, which related to specific genomic regions like simple repeats and segmental duplications.

**Population frequency-based filter.** The population frequency of the SNV did not exceed 1% in any of the following population based database- 1000 G, EXAC or ESP6500, according to the annotation results of Annovar.

**Force calling.** Multi-regional sequencing allowed the opportunity to increase the sensitivity to detect variants with low frequency. For a somatic mutation that was not detected ubiquitously across all tumour regions in a patient, reads information was extracted from corresponding bam files of each region which with a negative call, by using bam-readcount (0.8.0)[12]. In such cases, if mapping quality >20 and VAF >2%, this site was treated as a positive call.

**Indel filtering.** For InDel filtering, the blacklist filter and population frequency-based filter were the same as described above. VAF threshold was set to 5% for VarScan2 with somatic $P$ value $\leq 0.05$, total read depth was set >50, alternative read depth >10 in tumour samples and <2 in corresponding normal samples.

## CNA calling

ASCAT was used to estimate somatic copy number alternations (SCNA) of paired tumour-normal sequencing data. Allele counts of positions from 1000 genomes were generated using AlleleCounter, and minimum coverage of 20 for normal sample was used for filtration. LogR and BAF values were produced for each region, and concatenated into one matrix separately for each patient. LogR values were subsequently corrected using a GC wave correction implemented in ASCAT, and only heterozygous BAF values were reserved for further analysis. Allele-specific segmentation was performed to generate segmented logR and BAF data by ascat.aspcf. Manual verification was used to select the optimal model for ploidy and cellularity using an orthogonal measures based on ABSOLUTE results and mutation variant allele fraction. And then ASCAT was re-run to obtain the final allele-specific copy number data using reviewed cellularity and ploidy.

## HRD signature analysis

HRD scores are determined using the scarHRD R package. HRD score based on allele-specific copy numbers is sum of loss off heterozygosity (LOH), telomeric allelic imbalance (TAI), large-scale transitions (LST) scores. HRD-LOH score is the number of 15 Mb exceeding LOH regions which do not cover the whole chromosome. HRD-TAI is allelic imbalances that extend to the telomeric end of a chromosome. HRD-LST is defined as chromosomal break between adjacent regions of at least

10 Mb, with a distance between them not larger than 3 Mb. To examine specific (DDR) genes a curated geneset comprising homologous recombination DNA repair associated with biallelic inactivation in a pan-cancer meta-analysis was used.

Random forest classification was used to select relevant DDR genes differing significantly between R- and NR-subgroups. The McNemar's statistical test was performed for each variable comparing its importance with the maximum value of all random, also called shadow variables. Variables ($x$-axis) with significantly larger or smaller importance ($y$-axis) were classified/coded as important (green), unimportant (red), respectively, and indicated by colour-coding. Analysis was repeated 10 times using 5000 iterations each. The R statistical software version 4.1 and the'ranger' package were used for random forest training and variable importance elimination.

## HLA typing and HLA-LOH

HLA typing for MHC class-I genes was carried out using POLYSOLVER(v1.0) software for all 28 normal-samples' bam files, with default parameters. In brief, reads in the WES data potentially originate from HLA gene region were extracted out and then aligned to genomic sequence library of all known HLA alleles based on IMGT, using Novoalign packaged in POLYSOLVER. After which, a two-step Bayesian classification approach was used to infer the two alleles for each HLA class-I genes (HLA-A, HLA-B and HLA-C). A crucial part of neoantigen presentation is the HLA class-I genes products, which can present tumour associated epitopes to T-cells and then trigger out immunal response of body.

Loss of heterozygosity in HLA genes may lead to decreased ability to present productive tumour neoantigens, which could facilitate immune evasion of cancer. LOHHLA software was used to evaluate HLA loss for all 118 tumour samples, based on the alignment results of both tumour and corresponding normal samples, inferred tumour purity and ploidy information, and the HLA class-I genotyping results detected above. In brief, HLA reads were extracted and re-aligned to the patient-specific HLA-I alleles, then HLA gene specific log ratio was calculated based on coverage information on mismatch positions between homologous HLA alleles, and finally, HLA haplotype specific copy number was determined. In the analysis, items with PVal_unique $\leq 0.01$ (difference in log ratio between allele 1 and allele 2 $\leq 0.01$) were considered as a LOH event.

## Neoantigen prediction

In this analysis, neoantigens were defined as 8-11-mer peptides resulted from somatic SNVs or InDels which lead to amino-acid changes and, binding affinity score between remodelled peptide and respective patient's HLA class-I molecules was <500 nM. Somatic mutation VCF files both from VarScan2 and Mutect2 were annotated by Variant Effect Predictor (Version 84) with default parameter, except for the using of 'downstream' and 'wild-type' plugins offer by pVACseq. After annotation, the variants items lead to peptide changes were extracted out for downstream analysis. Bam-readcount (0.8.0) was used to acquire sequencing-based read depth information on each selected variant for both tumour and matched normal samples. Annotated non-synonymous mutations, sequencing-based information as well as HLA class-I gene typing results inferred by POLYSOLVER were feed into pVACseq(4.0.9) for neoantigen prediction. For each pVACseq run, epitope prediction was done by both NetMHC and NetMHCpan algorithms packed in pVACseq toolkit, epitope length was set to 8–11 and tumour DNA VAF cutoff was set to 10, with default parameters used for all other settings. Epitope prediction was performed based on the selected prediction algorithms, after which, sequencing-based information was integrated to enable filtering of neoantigen candidates (Normal Coverage $\geq 5X$, Normal VAF $\leq 2\%$, Tumour Coverage $\geq 10X$, Tumour VAF $\geq 40\%$). Inferred neoantigen candidates were selected out and those with binding affinity fold change >2 were considered with

higher priority level, which means the ratio of binding affinity score between wild-type peptide and mutated peptide. The greater this value, the stronger of the binding affinity after mutation compared with wild-type epitope.

## RNA sequencing

A Biomedical Scientist identified and marked representative areas of tumour on H&E stained slides from FFPE tumour blocks. Multiple tissue cores (1.0 mm each in size) were taken from the marked areas. RNA was isolated from these tissue cores using the MagMAXTM FFPE DNA/RNA Ultra Kit (ThermoFisher Scientific, Waltham, MA, USA #A31881) on the KingfisherTM Flex sample purification system (ThermoFisher Scientific, Waltham, MA, USA) as per manufacturer's instructions. RNA was quantified using the Qubit™ RNA HS Assay kit (ThermoFisher Scientific, Waltham, MA, USA Q32852) on the Qubit™ 4.0 fluorometer (ThermoFisher Scientific, Waltham, MA, USA) according to manufacturer's instructions.

A total amount of 2 μg RNA per sample was used as input material for the RNA sample preparations. Sequencing libraries were generated using NEBNext® UltraTM RNA Library Prep Kit for Illumina® (NEB, Ipswich, MA, USA) following manufacturer's recommendations and index codes were added to attribute sequences to each sample. Briefly, mRNA was purified from total RNA using poly-T oligo-attached magnetic beads. After fragmentation, the first strand cDNA was synthesised using random hexamer primer followed by the second strand cDNA synthesis using dTTP. Remaining overhangs were converted into blunt ends via exonuclease/polymerase activities. After adenylation of 3′ ends of DNA fragments, NEBNext Adaptor with hairpin loop structure were ligated to prepare for hybridisation. To select cDNA fragments of preferentially 150 ~ 200 bp in length, the library fragments were purified with AMPure XP system (Beckman Coulter, Brea, CA, USA). The concentration of each library was measured with real-time PCR.

Pools of the indexed library were prepared for cluster generation and PE150 sequencing on an Illumina NovaSeq 6000 (Illumina, San Diego, CA, USA). Fastp (0.12.2) was used to remove low-quality reads and reads containing sequencing adaptors. The processed reads were aligned using STAR (2.6.1 d) onto the human genome reference (UCSC hg19), and the transcripts were annotated based on gencode V19 gene models. Only the reads unique to one gene and which corresponded exactly to one gene structure were assigned to the corresponding genes by using HTSeq Counts were normalised for library size using estimateSizeFactors in Deseq2. FPKM data were generated using the fpkm function in Deseq2.

## Gene re-arrangements

STAR aligner (v.2.7.9a) was employed to produce chimeric alignments from RNAseq data, utilising a reference genome index file. These chimeric alignments are saved in a BAM file, which is then utilised by Arriba (v.2.1.0) to detect fusion events through the analysis of split and paired-end (discordant) read mapping. To ensure specificity, fusion predictions were filtered for somatic fusions by excluding recurrent artifacts, such as fusions originating from read-throughs, noncanonical splicing, and internal tandem duplication. STAR-Fusion (1.9.0) was applied as a second method to predict gene fusion events from RNAseq data.

## Immune repertoire analysis

We applied TRUST4 (v1.0.0) to obtain TCR and BCR clonotypes from bulk-RNAseq data for each sample. Raw pair-end reads were aligned to hg19 TCR/BCR sequences, and candidate reads were extracted to perform de novo assembly on V, J, C genes including the hypervariable complementarity determining region 3 (CDR3). The assembled consensus sequences were re-aligned to IMGT reference gene sequences for annotation. The statistics of TCR and BCR, including abundance, richness, shannon entropy and clonality, were compared between

reduction and no reduction groups using the Kruskal-Wallis rank-sum test.

## Gene set enrichment analysis

The gene set enrichment analysis (GSEA) was performed by applying the fGSEA R package. Row read counts were normalised and all genes ranked according to Wald test $P$ value by using the DESeq2 R package. For multiple correction, the false discovery rate (FDR) approach of Benjamini−Hochberg adjusted $P$ value was applied, and < 0.05 was considered significant. The visualisation of the results was conducted with the help of the ggplot2 R package.

## Immune cell fraction deconvolution

To infer immune cell fractions utilised four immune deconvolution tools: quantiseq, EPIC, MCP counter CIBERSORTx using bulk RNAseq data. Next, ground truth data was used (microscopy counts against deconvolution score) to investigate highly significant immune signatures found in CONFIRM cohort.

## Quantification of ERV expression

ERVmap was applied to determine the transcription of human endogenous retroviruses in RNA sequencing data. Briefly, RNA-seq reads were aligned against the ERV sequence library and human GRCh38 reference respectively. ERV reads were counted using ht-seq tools, and normalised by the total number of aligned RNA-seq reads in DEseq.

## Multiplex immunofluorescence

FFPE sections from baseline (pre-treatment) biopsies were deparaffinised and rehydrated using standard procedures. For heat-induced epitope retrieval sections were microwaved in 10 mM Tris/1 mM EDTA (pH 9.0) for five minutes, followed by 15 minutes at 30% power. Epitope-retrieved sections were mounted onto Sequenza hydrophobic clips (ThermoFisher Scientific, Waltham, MA, USA) and stained using an Opal 6-Plex Manual Detection Kit (Akoya Biosciences, Marlborough, MA, USA) according to manufacturer's instructions.

Briefly, sections were blocked with 1x Antibody Diluent/Block for 10 minutes, and stained with primary antibodies (diluted in PBS) for 30 minutes at room temperature, followed by secondary incubation with 1x Opal Anti-Ms+Rb HRP polymer for 30 minutes.

Primary Antibodies and opal fluorophores are shown below

| Marker | Antibody clone | Antibody dilution | Supplier | Paired Fluorophore |
|--------|----------------|-------------------|----------|--------------------|
| CD8 | C8/144B | 1:200 | Dako | Opal 480 (1:150) |
| CD4 | 4B12 | 1:50 | Dako | Opal 520 (1:200) |
| TIM-3 | D5D5r | 1:200 | Cell Signalling Technology | Opal 570 (1:250) |
| TIGIT | E5Y1W | 1:50 | Cell Signalling Technology | Opal 620 (1:250) |
| PD1 | EH33 | 1:200 | Cell Signalling Technology | Opal 690 (1:300) |
| CD19 | EPR5906 | 1:300 | Abcam | Opal 780 (1:50) |

Fluorescence signals were developed by 10-minute incubation with Opal fluorophore (480, 520, 570, 620, or 690) in 1x Plus Amplification Diluent. Multiplexing was achieved by iterating this process for each primary antibody/Opal fluorophore pair. For the final (sixth) round of staining, Opal TSA-DIG was used instead of fluorophores, followed by Opal 780 fluorophore incubation (1:50 dilution in 1x Antibody Diluent/Block) for 60 minutes. Slides were

then counterstained with 4′6-diamidino-2-phenylindole (DAPI, 6 µM) for 5 minutes and mounted using ProLong™ Diamond mounting media (ThermoFisher Scientific, Waltham, MA, USA). Sections similarly treated with omission of fluorophore/DAPI incubation were used for auto-fluorescence compensation in downstream image processing.

## Image analysis

Whole slide scanning was performed using a Vectra Polaris™ (Akoya Biosciences, Marlborough, MA, USA) automated quantitative pathology imaging system (multispectral slide scan mode with 0.50µm pixel resolution), according to manufacturer's instructions. Acquired whole scan image files were imported into inForm 2.6.0 image analysis software (Akoya Biosciences, Marlborough, MA, USA), and quantitative image analysis was performed with following steps:

(1) auto-fluorescence compensation;
(2) tissue segmentation into TLS/non-TLS areas based on CD19/CD4/DAPI/autofluorescence parameters;
(3) cell (nuclear) segmentation based on DAPI staining;
(4) single-cell phenotyping based on multiplex marker staining.

Tissue/cell segmentation algorithms were trained using >15 independent mesothelioma tissues obtained prior to this study, and the trained algorithms were further fine-tuned with each image file used in this study. To calculate the percentage of the cells with each phenotype, automated cell phenotyping/counting using the fine-tuned algorithms was performed throughout the tissue in an unbiased manner, and single-cell data-outputs containing not only phenotypes but positional and fluorescence intensity information were compiled using Python (version 3.6, package pandas 1.1.5) or R (version 4.2.3). The numbers and percentage of the cells with each phenotype was determined for each patient and densities of phenotypes in analysed tissues were calculated. Additionally, % of TLS areas were calculated as well as densities of $CD19^+$ and $CD8^+$ lymphocytes in TLS areas.

Random forest-based feature selection analysis was run on the percentages of the cells to identify the cell phenotypes best correlated with tumour response (Boruta_py 0.3, 5000 maximum iteration, $P$ value threshold 0.05). Features indicated by Boruta algorithm were further tested using the Wilcoxon rank sum test, unpaired, 2 sided with $P$ value threshold of 0.05 for significance (Graphpad prism 9.4.1). CD4 and CD8 markers were selected as important feature for further testing regardless of selection done by Boruta.

## Reporting summary

Further information on research design is available in the Nature Portfolio Reporting Summary linked to this article.

## Data availability

Trial data relating to this publication shall remain confidential to the sponsor organisation and will not be disclosed, except when disclosure might be required in accordance with pharmacovigilance duties of the parties involved. Individual participant data can be made available, after deidentification, to investigators who provide a written request in accordance with General Data Protection Regulation and following authorisation from the sponsor organisation, starting immediately and ending 3 years after publication. Data sharing requests should be directed to D.A.F and G.O.G. Southampton Clinical Trials Unit (SCTU), University of Southampton, Southampton, UK, is committed to the responsible sharing of clinical trial data and trial samples with the wider research community. Data access is administered through the SCTU Data Release Committee. Requests for data access and sharing for SCTU trials should be emailed to the SCTU Data Release Committee Coordinator at ctu@soton.ac.uk. The WES and RNA-sequencing raw data is available in SRA Run Selector. The data can be publicly accessed upon publication via (https://www.ncbi.nlm.nih.gov/bioproject/ PRJNA1148791), which is hosted by the National Centre for Biotechnology Information, under accession number PRJNA916814. All of the other data supporting the findings of this study are available within the article and its supplementary information files and from the corresponding author upon reasonable request. Source data are provided with this paper.

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

## Acknowledgements

We would like to thank the CONFIRM investigators who supported this clinical trial (listed in the supplementary information). Funding was provided by Stand Up to Cancer/Cancer Research UK (C16728/A21400). Bristol Myres Squibb (CA 209-841) supplied the study drug, its labelling and distribution with additional translational research funding support. Research was carried out at the National Institute for Health and Care Research (NIHR) Leicester Biomedical Research Centre (BRC). The views expressed are those of the author(s) and not necessarily those of the NIHR or the Department of Health and Social Care. J.C.H. was supported by the National Institutes of Health and Care Research Biomedical Institute, Leicester. J.S. was supported by the Medical Research Council. We thank the NIHR Leicester Biomedical Research Centre, NIHR Southampton Biomedical Research Centre, Cancer Research UK Experimental Cancer Research Centre, and University of Leicester advanced imaging facility for their support. We are most grateful to the patients who participated in this clinical trial, their families, and the nursing and medical staff at the CONFIRM trial sites. We would also like to thank Mesothelioma UK, the independent clinical trial steering committee, and Bristol Myers Squibb for the provision of nivolumab. We would further like to acknowledge the guidance the members of the CONFIRM independent data monitoring committee (Prof Pieter Postmus, Prof Sanjay Popat and Andre Lopes) have offered us throughout the duration of the trial. This study was sponsored by the University of Southampton.

## Author contributions

D.A.F. conceptualised the study and secured study funding from Cancer Research UK and Standup to Cancer. D.A.F., S.E., K.H., and G.O.G. developed the clinical methods. SE, K.H., K.M., L.J., and J.L. did the formal clinical analysis and validated the underlying data. C.M. and L.J. managed project administration. M.N. was part of the trial management group and was the patient representative. D.A.F, C.O, R.C, G.G.H, N.S, P.S, S.C, L.D., S.D., J.F.L recruited participants. P.W.J., C.P., and C.R. curated the conducted the PD-L1 analysis. C.P., M.Z, E.Y.B, J.D, N.N., J.R., A.B., J.H., J.S., D.F., C.Prichard, T.K., J.C.H., M.Jama, E.H., J.L.L., Z.Z., H.Y., H.Z., A.K., we are involved in planning, conducting, and interpreting multi-omic analyses. All authors wrote, reviewed, edited the final draft, and had final responsibility for the decision to submit for publication. DAF and GG verified the data. All authors had full access to all of the data in the study.

## Competing interests

DAF reports grants from Aldeyra, Astex Therapeutics, Bayer, Bergen Bio, Boehringer Ingelheim, Bristol Myers Squibb, GSK, Iovance, Merck Sharp & Dohme, Owkin, RS Oncology; non-financial support from Clovis, Eli Lilly, Roche, and Bristol Myers Squibb. Personal fees from Astra Zeneca, Boehringer Ingelheim, Bristol Myers Squibb, GSK, Orion, RS Oncology and non-financial support from Roche. GG reports grants from Jannsen-Cilag, Novartis, Astex, Roche, Heartflow, Bristol Myers Squibb, BioNtech; grants and personal fees from AstraZeneca; and personal fees from Celldex, outside the submitted work. JL reports grants from Cancer Research UK and non-financial support from Bristol Myers Squibb. CO reports personal fees from Bristol Myers Squibb. RC reports personal fees from Bristol Myers Squibb, Merck Sharp & Dohme, Roche, and AstraZeneca. All other authors declare no competing interests.

## Additional information

Dean A. Fennell [1,2] ✉, Kayleigh Hill [3], Min Zhang[1,4], Charlotte Poile[1], Sean Ewings[3], Essa Y. Baitei[1,5], Joanna Dzialo [1], Nada Nusrat[1,6], Jan Rogel[1], Daniel Faulkner [1], Christian Ottensmeier [7], Raffaele Califano[8], Gerard G. Hanna [9], Sarah Danson[10], Nicola Steele[11], Mavis Nye[12], Lucy Johnson[3], Kim Mallard[3], Joanne Lord[3], Calley Middleton[3], Peter Szlosarek[13], Sam Chan[14], Liz Darlison[15], Peter Wells-Jordan[1,2], Cathy Richards[2], James Harber [16], Aleksandra Bzura[1], Jake Spicer[1], Catrin Pritchard[1], Tamihiro Kamata[17], Jens C. Hahne[1], Maymun Jama[1], Edward J. Hollox [18], Jason F. Lester[19], Jin-Li Luo[20], Zisen Zhou [21], Hongji Yang [21], Huiyu Zhou[21], Astero Klampatsa [22] & Gareth O. Griffiths [3]

[1]Mesothelioma Research Programme, CRUK Experimental Cancer Centre & NIHR Biomedical Research Centre, University of Leicester, UK Robert Kilpatrick Clinical Sciences Building, Leicester, UK. [2]Department of Oncology, University Hospitals of Leicester NHS Trust, Leicester, UK. [3]Cancer Research UK Southampton Clinical Trials Unit & NIHR Southampton Biomedical Research Centre UK, University of Southampton, Southampton, UK. [4]Novogene Co. Limited, Beijing, China. [5]Center for Genomic Medicine, King Faisal Specialist Hospital & Research Centre, Riyadh, Saudi Arabia. [6]Faculty of Medicine, University of Tripoli, Tripoli, Libya. [7]Department of Molecular & Clinical Cancer Medicine, University of Liverpool, Liverpool, UK. [8]Department of Medical Oncology, Wythenshawe Hospital, Manchester, UK. [9]Patrick G. Johnston Centre for Cancer Research, Queens University Belfast, Northern Ireland, UK. [10]Department of Oncology, Sheffield Teaching Hospitals NHS Foundation, Sheffield, UK. [11]Beatson West of Scotland Cancer Centre, Glasgow, Scotland, UK. [12]Mavis Nye Foundation, c/o University of Southampton, Southampton, UK. [13]Cancer Research UK Barts Cancer Institute, Queen Mary University of London, London, UK. [14]York Teaching Hospital NHS Foundation Trust, York, UK. [15]Mesothelioma UK, Loughborough, UK. [16]UWA Centre for Medical Research & Harry Perkins Institute of Medical Research, Perth, Australia. [17]MRC Toxicology Unit, University of Cambridge, Gleeson Building, Tennis Court Road, Cambridge, UK. [18]Department of Genetics and Genome Biology, University of Leicester, Leicester, UK. [19]Singleton & Morrison Hospitals, South Wales, UK. [20]Bioinformatics Analysis Support Hub, University of Leicester, Leicester, UK. [21]School of Computing and Mathematical Sciences, University of Leicester, Leicester, UK. [22]Thoracic Oncology Immunotherapy Group, The Institute of Cancer Research, Sutton, London, UK. ✉e-mail: df132@le.ac.uk

