## [Transparent Peer Review file · Nature Communications]

Constitutive Inflammation and Epithelial-Mesenchymal Transition Dictate Sensitivity to Nivolumab in CONFIRM: a placebo-controlled, randomised phase III trial

Corresponding Author: Professor Dean Fennell

Version 0:

Reviewer comments:

Reviewer #1

(Remarks to the Author)

The authors present the final results from the CONFIRM trial, which was a placebo controlled double-blind randomised phase III trial to assess the efficacy of Nivolumab in patients with relapsing mesothelioma.

The trial enrolled 332 participants and 221 were randomised 1:1.

The primary endpoints were met with longer PFS in the Nivolumab arm.

This is an important study considering the lack of options for second-line therapy in mesothelioma.

The authors performed a multi-omics analysis to determine markers associated with response to Nivolumab in a small subgroup of 23 patients, including 10 responders and 13 nonresponders. Supplementary table 1D mentioned 26 patients with 16 responders, but the difference in number in the R group is unclear.

The analysis demonstrated that CD8 enrichment and TLS were associated with response and EMT with lack of response. The authors performed an independent study showing that these factors were not associated with PFS after pleurectomy-decortication.

The analysis unfortunately included a small population of patients and while the results are supported by other publications and mechanistic studies, it remains unclear if the observation is an association or a direct mechanistic link. It is also unclear if the biopsies used for the analysis were performed before or after first-line therapy, which may have affected the CD8 T cell populations or EMT process.

Reviewer #2

(Remarks to the Author)

The authors have addressed my comments and revised the manuscript. I have some minor comments based on the revised manuscript for their consideration.

1. The added information on survival is incomplete and leaves out the hazard ratio at line 136.
2. Lines 147-150 should be updated to include Figure 1A with OS.
3. Although I agree that STAR-fusion and Arriba are validated methods for calling gene fusions, the issue is the use of FFPE which degrades DNA length and reduces the sensitivity of any method for calling fusions. This should be a limitation that is acknowledged.
4. Lines 251-252 have redundancies as written.

5. Figure 1 legend should be updated to include OS, and labeled for clarity.

Reviewer #3

(Remarks to the Author)

The authors studied the factors associated with the responsiveness to PD-L1 blockade in a phase III clinical trial of mesothelioma. They found that pro-inflammatory transcription and TLSs were enriched in nivolumab responders, while EMT and mitosis were associated with resistance to nivolumab. These results, though expected, are valuable for immunotherapy in mesothelioma. After revision, the current manuscript is qualified for publication in this journal.

Reviewer #4

(Remarks to the Author)

I believe the authors have adequately addressed the comments from all reviewers through text clarifications, figure changes, and additional cross-validation cohorts.

Reviewer #5

(Remarks to the Author)

I have reviewed the manuscript, "Constitutive Inflammation and Epithelial-Mesenchymal Transition Dictate Sensitivity to Programmed Death-1 Inhibition in Mesothelioma" and the response letter. The authors reported adjusted hazard ratios of 0.65 (95% CI: 0.51, 0.82, p-value < .001) and 0.81 (95% CI: 0.64, 1.04, p-value = 0.096) in favor of Nivolumab for PFS and OS. Only the efficacy endpoint of PFS was met. The comments raised by reviewer 2 except 2.10 and 2.12 are adequately addressed.

The reviewer 2's comments 2.10 and 2.12 about the survival analysis between responders and non-responders were not addressed. At the time of randomization, the best response status (R and NR) was unknown. The status was known when CT was performed at evaluation. The appropriate way to perform survival analysis by response status is to use the best response status date as the starting timepoint. A patient could have no response in first evaluation and had a response in second evaluation, the starting timepoint for this patient was second evaluation. A patient could have no response in first evaluation and remained no response, the starting time point for this patient was first evaluation.

Minor comments that can help the reading of the manuscript:

1. In supplementary table 1, PS could be added after ECOG. 0 (0.0)% could be added in the Missing from eCFR row. In the title, please capitalize Confirm. Please check the numbers of PD-L1 positive, negative, and missing in this table as these numbers are different from the numbers published previously in Table 1 in Fennell et al. 2021?
2. Please provide the data cutoff date for the survival outcome.
3. In supplementary table 4, please add the unit month in median progression free survival.
4. In supplementary table 5, please change overall survival to progression free survival and add the unit month.
5. In supplementary figures 2 and 4, the p-value of asbestos exposure is missing. In supplementary figure 2, the unadjusted p-value comparing PFS between Nivolumab and placebo was not correct. Should it be < .001?
6. In supplementary Table 1, the PD-L1 categories 1 – 49% and > 50% are used. However, in supplementary figures 2, 3, and 6, 1 – 50% is used. In supplementary figure 3, please change the title in each panel from PDL1 to PD-L1. In supplementary figure 4, >=50% is used. In supplementary figure 6 title, please change PD-L1 > 1 % to PD-L1 > 1 – 49%. Please be consistent in PD-L1 categories.
7. In line 145, 0.28 (0.09-0.94, p = 0.04) is from supplementary table 5 but it is not referred.
8. In line 149, it says supplementary figure 5. Should it be Figure 1A?
9. In line 154, it says supplementary figure 7A-C. Should it be supplementary figure 6A-C?
10. In line 157-158, it says the odds ratio is 12.7 (95% CI, 2.1-undefined limit, p= 0.002). Is the odds ratio calculated from ITT population or the population evaluable for modified RECIST? The upper bound of the confidence interval should be able to calculate.
11. Supplementary table 8A is not mentioned in the manuscript. Please change Progression to Progressive disease. Please check the numbers of response in supplementary table 8A, B as these numbers are different from the numbers published previously in Table 2 in Fennell et al. 2021?
13. In line 167-170, it says leading to treatment discontinuation in the nivolumab arm were infusion related reactions, ... (supplementary table 10). But supplementary table 10 does not mention treatment discontinuation.
13. Number of responders and non-responders should be mentioned in the multi-omic analysis in the manuscript.
14. In line 185, it says supplementary figure 6. Should it be supplementary figure 5C? Supplementary figure 5A and 5B are not referred.
15. In Figure 1F and G, landmark analysis should be performed per reviewer 2's comment. The label of x-axis should say time from best response.
16. In line 197, there is no supplementary figure 8D.
17. In line 200, it says supplementary figure 8A-C. Should it be supplementary figure 7A-C?
18. In line 204, it says supplementary figure 9B-C. Should it be supplementary figure 8B-C? Supplementary figure 8A is not referred.
19. In line 209, should it be supplementary figure 9? PFS in the title should be removed.
20. In line 220-221, it says CIBERSORTx Wilcoxon p=0.029, MCP Counter, Mann Whitney U test p = 0.017 figure 3G. Wilcoxon and Mann Whitney U test are the same test. Please use the same term for the test.

21. In Figure 3I middle and right panels, the label of the ticks on x-axis do not seem correct. The curves ends at around 2250 days in these two panels but the curve in the left panel ends at around 1300 days.
22. In line 229, it says supplementary figure 12. Should it be supplementary figure 10?
23. In line 234, it says supplementary figure 13. Should it be supplementary figure 11? Figure 4A and Figure 4G are not referred in the manuscript.
24. In Figure 1A legend, overall survival should be added.
25. In the statistical analysis method section, please add the statistical tests that were used and the landmark analysis.

Version 1:

Reviewer comments:

Reviewer #2

(Remarks to the Author)

I am satisfied with the authors' responses to the comments. I have no further questions or concerns.

Reviewer #5

(Remarks to the Author)

I do not have further comment about the landmark analysis. I still have some comments after reading the revised manuscript.

1. Only PFS is shown in Figure 1A. Please add OS.
2. Figure 1E is not referred in the manuscript.
3. Please change "Progression free survival" to "Overall survival" in the title in Supplementary Table 7.
4. In line 160, it says Supplementary Figure 5A-C. But Supplementary Figure 5A-C are the RECIST response, spider plot, and heatmap which are not related to overall survival. I think Supplementary Figure 6 should be Supplementary Figure 5 and vice versa.
5. Comment 11 regarding the discrepancy in objective response is not addressed. In Supplementary Table 8 in this manuscript, there are 22 partial responses. In Table 2 in the article Fennell et al. 2021, there are 25 partial responses. If a patient had partial response as best overall response in the 2021 article, this patient could not have a different best overall response in this article given there is no complete response. The number of partial responses can only increase from 2021. Please check the number of responses.

Reviewer #1:

Reviewer's comment

1.1 Supplementary table 1D mentioned 26 patients with 16 responders, but the difference in number in the R group is unclear.

Authors' response

There is no supplementary table 1D in the manuscript, only supplementary tables 1A-C which are referred. To address the size of the multi-omics cohort, this is now added in the revised manuscript ; R, 16 patients and NR, 13 patients. In addition there reference to a heatmap in supplementary figure 5C which is referred to showing the specific multi-omic studies undertaken in this cohort.

Reviewer's comment

1.2 The analysis unfortunately included a small population of patients and while the results are

supported by other publications and mechanistic studies, it remains unclear if the observation is an

association or a direct mechanistic link.

Authors' response

Agreed. A statement has been included in the discussion reflecting this limitation of a small sample size in the revised manuscript.

Reviewer's comment

1.3 It is also unclear if the biopsies used for the analysis were performed before or after first-line therapy, which may have affected the CD8 T cell populations or EMT

process.

Authors' response

To provide further clarification (the term diagnostic tissue blocks is used at line 191 in the revised manuscript, with the inclusion of “prior to first-line therapy” also added at line 191 to remove any ambiguity.

Reviewer #2:

Reviewer's comment

2.1 The added information on survival is incomplete and leaves out the hazard ratio at line 136.

Authors' response

The hazard ratio for survival is now added in the revised manuscript.

Reviewer's comment

2.2 Lines 147-150 should be updated to include Figure 1A with OS.

Authors' response

Agreed. Figure 1A is now mentioned

Reviewer's comment

2.3 Although I agree that STAR-fusion and Arriba are validated methods for calling gene fusions, the

issue is the use of FFPE which degrades DNA length and reduces the sensitivity of any method for

calling fusions. This should be a limitation that is acknowledged.

Authors' response

I believe the reviewer is referring to RNA degradation rather than DNA (as RNA sequencing data was used to call fusions). We agree with this limitation however and a comment has been added.

Reviewer's comment

2.4 Lines 251-252 have redundancies as written.

Authors' response

Agreed – the repeated term “ in the final analysis” has been deleted in the discussion.

Reviewer's comment

2.5 Figure 1 legend should be updated to include OS, and labeled for clarity.

Authors' response

Agreed. The figure legend (1G) has been amended to include reference to the overall survival Kaplan Meier plot in the lower panel.

Reviewer #5 (Biostatistical analysis):

Reviewer's comment

5.1 The reviewer 2's comments 2.10 and 2.12 about the survival analysis between responders and non-responders were not addressed. At the time of randomization, the best response status (R and NR) was unknown. The status was known when CT was performed at evaluation. The appropriate way to perform survival analysis by

response status is to use the best response status date as the starting timepoint. A patient could have no response in first evaluation and had a response in second evaluation, the starting timepoint for this patient was second evaluation. A patient could have no response in first evaluation and remained no response, the starting time point for this patient was first evaluation.

Authors' response

Thank you for this comment. Our statisticians have given this important comment careful consideration leading to essential amendments which are included in the revised manuscript.

The primary objective of these plots was to provide descriptive information only on the translational cohort, ie. show the characteristics of, but not to formally compare the OS and PFS of patients in R and NR groups undergoing multiomic analysis. We accept that the rules of landmark analysis are required when formally comparing response versus survival (eg. Fennell et al, Eur J Cancer 2012 48(16) 2983 and and Hasan et al, Eur J Cancer 2014 50(16), 2771).

By definition the translational cohort comprising the extremities of response phenotype, required evaluation by modified RECIST to have occurred enabling groupings as either R (stable or reduction in tumour size as best response at any timepoint) or NR (growth as best response at any timepoint). We accept that the overlay of R and NR groups would prompt a comparative interpretation.

To address this, the original figure legends have been modified to avoid any ambiguity as follows.

1. The terms responder and non-responder originally are time-independent assignments that relate to whether or not the best response (at any time) was growth (NR group) or reduction/stability (R) in the tumour.

2. a statement has been added to both legends in figure 1F and G and the corresponding results section of the manuscript that the responder and non-responder groups are descriptive only and are not formally comparable.

Minor comments that can help the reading of the manuscript:

1. In supplementary table 1, PS could be added after ECOG. 0 (0.0)% could be added in the Missing

from eCFR row. In the title, please capitalize Confirm. Please check the numbers of PD-L1 positive, negative, and missing in this table as these numbers are different from the numbers published previously in Table 1 in Fennell et al. 2021². Please check the numbers of PD-L1 positive, negative, and missing in this table as these numbers are different from the numbers published previously in Table 1 in Fennell et al. 2021². Please provide the data cutoff date for the survival outcome.

Authors' response

The table has been updated accordingly – the PS has been added after ECOG, and the Confirm has been capitalized. It should be noted that PDL1 assay was conducted on a smaller subset of patients in the preliminary report of CONFIRM (Fennell et al, Lancet Oncology 2021) compared to this final analysis. As such the denominator has increased in the final cohort, explaining this difference. A statement to explain this is included.

2. Please provide the data cutoff date for the survival outcome.

Authors' response

The analysis of overall survival was planned to coincide with 291 events. Patients who did not experience a survival event at this threshold were censored at their completed End of Study date. A statement reflecting this is included in the results.

3. In supplementary table 4, please add the unit month in median progression free survival.

Authors' response

Month units have now been added to the supplementary table 4.

4. In supplementary table 5, please change overall survival to progression free survival and add the unit month.

Authors' response

In supplementary table 5, overall survival has been changed to progression free survival and the unit month has been included

5. In supplementary figures 2 and 4, the p-value of asbestos exposure is missing. In supplementary

figure 2, the unadjusted p-value comparing PFS between Nivolumab and placebo was not correct.

Should it be $< .001$?

Authors' response

These changes have been corrected with updated supplementary figures 2 and 4

6. In supplementary Table 1, the PD-L1 categories 1 – 49% and > 50% are used. However, in supplementary figures 2, 3, and 6, 1 – 50% is used. In supplementary figure 3, please change the title in each panel from PDL1 to PD-L1. In supplementary figure 4, >=50% is used. In supplementary figure 6 title, please change PD-L1 > 1 % to PD-L1 > 1 – 49%. Please be consistent in PD-L1 categories.

Authors' response

Agreed. The labels have been amended for consistency to show the correct categories PDL1 <1%, PDL1 1-49%, and PDL1>50% across all figures and tables.

7. In line 145, 0.28 (0.09-0.94, p = 0.04) is from supplementary table 5 but it is not referred.

Authors' response

Reference to supplementary table 5 is now added

8. In line 149, it says supplementary figure 5. Should it be Figure 1A?

Authors' response

Correct, this should be figure 1A and supplementary table 6. An amendment has been made accordingly

9. In line 154, it says supplementary figure 7A-C. Should it be supplementary figure 6A-C?

Authors' response

The correct figure is supplementary figure 5A-C. A correction has been made accordingly

10. In line 157-158, it says the odds ratio is 12.7 (95% CI, 2.1-undefined limit, $p=0.002$). Is the odds

ratio calculated from ITT population or the population evaluable for modified RECIST? The upper

bound of the confidence interval should be able to calculate.

Authors' response

The odds ratio corresponds to the population of patients who were evaluated by modified RECIST (305 patients). This statement has been added to the results.

11. Supplementary table 8A is not mentioned in the manuscript. Please change Progression to

Progressive disease. Please check the numbers of response in supplementary table 8A, B as these

numbers are different from the numbers published previously in Table 2 in Fennell et al. 2021?

Authors' response

Progression has been changed to progressive disease in supplementary tables 8A and 8B. The denominator in this final analysis is larger for the PDL1 assessed cohort as discussed above, and referred to in the revised manuscript.

12. In line 167-170, it says leading to treatment discontinuation in the nivolumab arm were infusion

related reactions, ... (supplementary table 10). But supplementary table 10 does not mention

treatment discontinuation.

Authors' response

Agreed. Reference to supplementary table 10 has been removed

13. Number of responders and non-responders should be mentioned in the multi-omic analysis in

the manuscript.

Authors' response

The total number of patients in the multiomic cohort is now included in the manuscript and is shown in supplementary figure 6C.

14. In line 185, it says supplementary figure 6. Should it be supplementary figure 5C?

Supplementary figure 5A and 5B are not referred.

Authors' response

Supplementary figures 5A-B are now referred to in the revised manuscript.

15. In Figure 1F and G, landmark analysis should be performed per reviewer 2's comment. The label

of x-axis should say time from best response.

Authors' response

As discussed above. In response to reviewer's 2 comment, we agree that landmark analysis would be needed if formally comparing response with PFS and survival, however the Kaplan Meier curves are purely descriptive for the corresponding PFS and OS in the translational research cohort, and as such shown as time from randomisation. The text has also been modified to highlight this presentation of the multi-omic cohorts PFS and OS characteristics.

16. In line 197, there is no supplementary figure 8D.

Authors' response

Correct, reference to figure 8D has been removed

17. In line 200, it says supplementary figure 8A-C. Should it be supplementary figure 7A-C?

Authors' response

Correct. The text has been modified accordingly

18. In line 204, i says supplementary figure 9B-C. Should it be supplementary figure 8B-C?

Authors' response

Correct. The text has been modified accordingly

Supplementary figure 8A is not referred.

Authors' response

Reference to supplementary figure 8A (fusions) is made in the revised manuscript.

19. In line 209, should it be supplementary figure 9? PFS in the title should be removed.

Authors' response

Correct. The figure has been amended to correct PFS to OS in the legend and the reference corrected to supplementary figure 9

20. In line 220-221, it says CIBERSORTx Wilcoxon $p=0.029$, MCP Counter, Mann Whitney U test $p =$

0.017 figure 3G. Wilcoxon and Mann Whitney U test are the same test. Please use the same term

for the test.

Authors' response

Thank you. The term Wilcoxon has been used for consistency throughout. In figure 3G we have recalculated the MCP counter score based on the more robust tpm normalised RNAseq data, which remains statistically significant for CD8+ T lymphocytes and is cross validated with multiplex immunofluorescence. In supplementary figure 8C this RNAseq based deconvolution result is cross validated using 3 different algorithms; quantiseq, EPIC and, Consensus TME. Please note MCP counter was duplicated in this supplementary figure and cibersort has been removed from the main figure to avoid clutter and because $p >.05$ on recalculation using tpm normalisation (amended in the methods at 1062).

21. In Figure 3I middle and right panels, the label of the ticks on x-axis do not seem correct. The

curves ends at around 2250 days in these two panels but the curve in the left panel ends at around

1300 days.

Authors' response

Agreed. The CD8 associated graph has had the X axis corrected to match the EMT and IL24 graphs with the maximum on the scale being adjusted to 2500.

22. In line 229, it says supplementary figure 12. Should it be supplementary figure 10?

Authors' response

Correct. This figure reference has been amended accordingly

23. In line 234, it says supplementary figure 13. Should it be supplementary figure 11? Figure 4A

and Figure 4G are not referred in the manuscript.

Authors' response

Agreed. Supplementary figure 13 has been corrected to supplementary figure 11

Reference to 4A is now included in the results text, and figure 4G (a summary schematic) has been added in the summary paragraph in the revised manuscript.

Figure 4G is mentioned at line 24. In Figure 1A legend, overall survival should be added.

Authors' response

Agreed. This has been amended accordingly. Reference to figure 4G is removed from the discussion and reference to overall survival added to the figure 1A legend.

25. In the statistical analysis method section, please add the statistical tests that were used and the

landmark analysis

Authors' response

Statistical tests related to the multi-omic cohort, including permutation testing for adjusted Wilcoxon p values has now been included in the statistical analysis section. The multi-omics cohort associated Kaplan Meier curves are descriptive for the R vs NR groups which are not formally compared as addressed above, and therefore landmark analysis is not described.

Reviewer's comment

1. Only PFS is shown in Figure 1A. Please add OS.

Author's response

This amendment has been made to the revised figure 1A

2. Figure 1E is not referred in the manuscript.

Author's response

Thank you. This typo has been corrected - the figure referred to is now figure 1C-E.

3. Please change "Progression free survival" to "Overall survival" in the title in Supplementary Table 7.

Author's response

Thank you. This change has been made accordingly

4. In line 160, it says Supplementary Figure 5A-C. But Supplementary Figure 5A-C are the RECIST response, spider plot, and heatmap which are not related to overall survival. I think Supplementary Figure 6 should be Supplementary Figure 5 and vice versa.

Author's response

Correct, supplementary figures 5 and 6 were reversed. The figure numbering has been re-ordered as highlighted.

5. Comment 11 regarding the discrepancy in objective response is not addressed. In Supplementary Table 8 in this manuscript, there are 22 partial responses. In Table 2 in the article Fennell et al. 2021, there are 25 partial responses. If a patient had partial response as best overall response in the 2021 article, this patient could not have a different best

overall response in this article given there is no complete response. The number of partial responses can only increase from 2021. Please check the number of responses.

Author's response

Thank you for highlighting this important point. In this final analysis of the clinical trial results, a centralised computation of the modified RECIST response outcome was conducted. Compared with the investigator reported responses reported in the preliminary results paper (2021, Lancet Oncology), this led to a reduction from 25% to 22%, accounting for the discrepancy noted. A statement explaining this difference is now added at line 168 to the manuscript:

It should be noted that the modified RECIST response rate was re-computed centrally by the clinical trials unit, resulting a reduction of 2% from the investigator-reported 25% rate reported in the preliminary results publication⁵